# Opportunities and Challenges for Cover Cropping in Sustainable Agriculture Systems in Southern Australia

Thomas Nordblom [1], Saliya Gurusinghe [1,*], Andrew Erbacher [2] and Leslie A. Weston [1]

[1] Gulbali Institute, Agriculture Water and the Environment, Charles Sturt University, Boorooma Street, Wagga Wagga, NSW 2678, Australia; tnordblom@csu.edu.au (T.N.); leweston@csu.edu.au (L.A.W.)
[2] Queensland Department of Agriculture and Fisheries, 26 Lagoon Street, Goondiwindi, QLD 4390, Australia
* Correspondence: sgurusinghe@csu.edu.au

**Abstract:** Southern Australian farming systems operate predominantly under Mediterranean climatic conditions, which limit the choice of cover crops suitable for enhancement of ground cover and soil moisture retention, erosion control, atmospheric soil nitrogen (N) fixation, and weed suppression between cash crop rotations. Given that the successful establishment of cover crops is climate-driven and also influenced by edaphic factors such as soil pH and salinity, there has been increased interest by southern Australian producers in identifying potential cover crop species well adapted to specific Australian farming systems, which provide vital ecosystem services and sustainable economic benefits through the improvement of soil properties. This review summarises recent findings on cover crop inclusion in diverse farming systems in southern Australia, including continuous and mixed broadacre cropping as well as viticulture and horticulture systems, to identify opportunities and limitations related to their use. Cover crop inclusion in viticulture and pasture systems with lower moisture stress was observed to benefit the subsequent cash crop through enhanced production potential. Long-term, multi-site field experimentation incorporating summer cover crops in winter crop rotations showed that cover crops enhanced ground cover and soil water infiltration in some locations across southern Australia while sometimes increasing winter crop yield, suggesting that soil type and regional climatic conditions greatly influenced the delivery of multiple cover crop benefits. Collectively, these studies have suggested a need for longer-term field evaluations using multiple cover crop species and investigations of termination options under varying environmental and soil conditions to better quantify the legacy effects of cover crops.

**Keywords:** cover crops; rotational crops; soil moisture conservation; economic benefits; sustainable cropping systems

## 1. Introduction

Conservation cropping systems exhibit relatively low weed densities (i.e., <1.0 plant m$^{-2}$) in Australia in recent years because growers have focused on minimizing weed populations while limiting the spread of herbicide resistance [1]. Low weed densities have been achieved through the careful employment of 'routine' weed-control options, which include efficient application of herbicide mixtures, employment of harvest weed-seed control systems such as the Harrington Seed Destructor, and use of crop competition for weed suppression [2–4]. In addition, 'strategic' weed control tactics to manage or reset weedy conditions include moldboard ploughing [5], the use of fallow cropping, and also the production of hay or silage [6,7]. Cover or smother crops [8] have also been successfully deployed when weed populations reach unacceptably high levels (e.g., >5.0 plants m$^{-2}$) [9].

Cover crops are used globally as they frequently provide important agro-ecological functions in various cropping systems by improving soil tilth and moisture availability for subsequent crops while suppressing annual weeds and reducing soil erosion [10,11]. Nitrogen-fixing legume cover crops may also enhance available N content in the rhizosphere [12], while providing fodder rich in nutritional value for livestock in mixed farming

systems [13]. The grain growing regions of Australia experience variable annual rainfall, with southern and western regions generally experiencing greater precipitation in winter months and northern regions experiencing more rainfall in summer months [14]. Encouraging infiltration and water storage in all Australian production systems is vital for the success of subsequent crops, as soil moisture frequently limits productivity across most regions. While monocot species such as sorghum (*Sorghum bicolor*), sudex (*Sorghum bicolor* × *Sorghum sudanese*), and millet (*Setaria* spp.) are commonly used as summer annual cover crops across Australia and oats (*Avena sativa* sp.) and pasture legumes as winter annual cover crops, concerns about the proliferation of weeds and grass-specific pathogens and nematodes through grass-grass crop rotations [15] suggest a critical need for greater cover crop species diversity in rotations implemented across the southern winter cropping regions [16,17].

Cover crops, when incorporated into integrated weed management strategies, have frequently resulted in reduced herbicide use and the incidence of herbicide resistance, both of which are important considerations for Australian grain producers [1,18]. The weed suppressive potential of cover crops typically manifests either through competition for resources [19] or by the release of phytotoxic secondary metabolites from crop residues and root exudates, resulting in inhibition of weed seed germination or a reduction in weed seedling growth [20,21]. However, the latter has not always been demonstrated under field conditions but is based on the concept that the cover crop and/or its dried residues release phytotoxins at concentrations in the soil rhizosphere sufficient to adversely impact weed seed germination and establishment directly or through their subsequent transformation to phytotoxins by associated soil microbiota [22].

Legume cover crops, which successfully fix N, and their residues have also been shown to effectively suppress weeds [23–25]. From the perspective of enhanced competition for resources, low-growing clovers and other densely established legume mixtures can provide adequate ground cover to suppress weed germination and establishment. Clover species are also known to be highly competitive and can reduce in-crop weed biomass significantly when compared to bare ground or cereal crop residues, and many have no appreciable yield penalties on successive crops [26,27]. Large-seeded legumes include the summer-planted Fabaceae and covers like velvet beans and cowpeas. These legumes have effectively inhibited the germination and emergence of annual grass and broadleaf weeds in successive field trials [28].

While certain cover crops have been shown to be weed suppressive, their incorporation into integrated weed management strategies has also resulted in a reduction in the frequency of herbicide applications in crop fallows in both southern and northern grain growing regions across Australia [29–31], thereby providing additional financial incentives for their inclusion. Cover crops are also used as living mulches for sustained productivity of irrigated horticultural crops, though they are not profitable in all locations [32,33].

This review summarises the potential role of cover crops in diverse farming systems encountered in southern Australia, including those containing continuous and mixed broadacre cropping as well as those featuring viticulture and horticulture crops, with the purpose of identifying opportunities and limitations related to their use patterns.

## 2. Cover Crop Species Selection: An Australian Perspective

Australian production systems can frequently support a diverse group of potential cover crops based on the following factors: seasonality desired (specifically, winter or summer sown), adaptation to diverse environments, early vigour, canopy architecture, root architecture, use patterns, as well as suitability for grazing livestock [2,34]. While Australia's land area suitable for crop production is estimated at less than 15% of its total landmass, those areas already in production contain diverse Mediterranean, tropical, subtropical, temperate, and desert regions, some equipped with suitable irrigation (Figure 1). The availability of irrigation or naturally occurring rainfall, along with suitably fertile soils

at an appropriate soil pH, are critical factors that determine if successful crop and rotational cover crop production will occur.

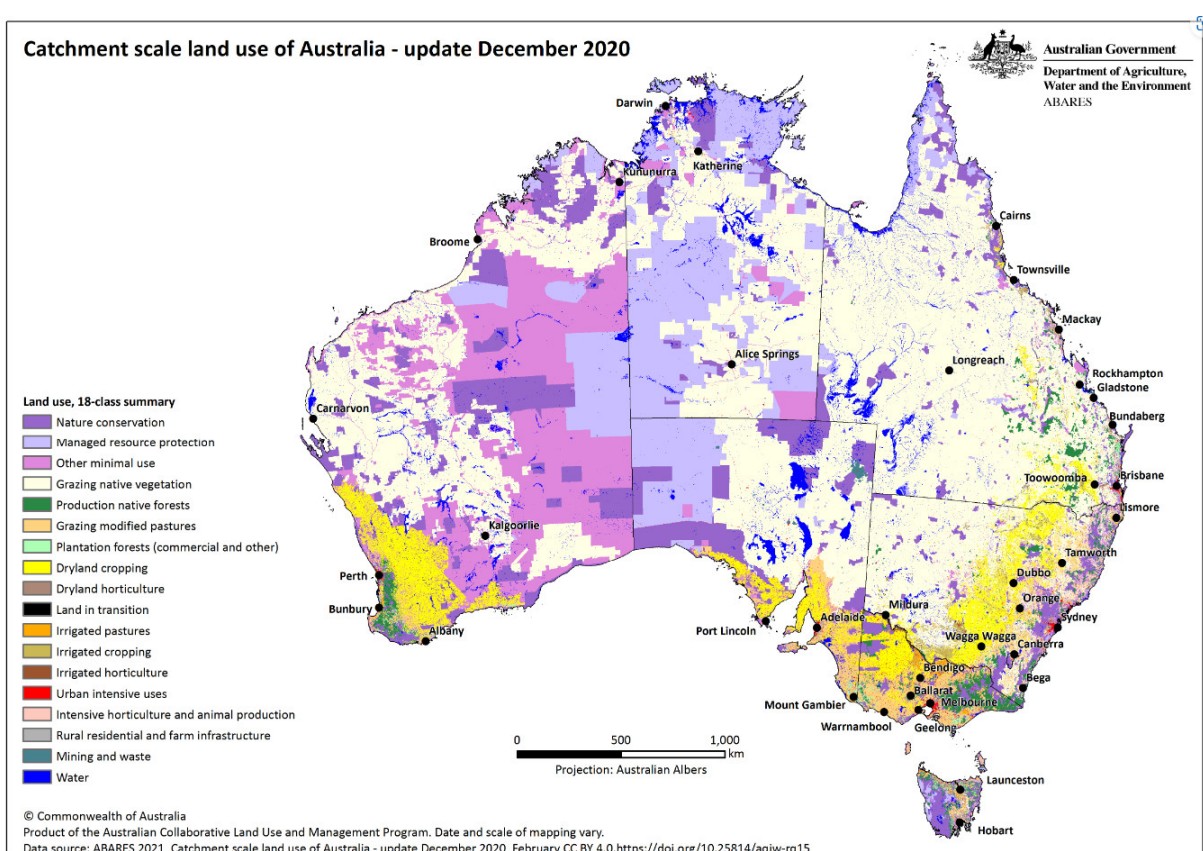

**Figure 1.** Land use categories in Australia. Cover crops are best utilised in four of the land use categories, including dryland and irrigated cropping and horticulture, represented in shades of yellow and brown. Image sourced from www.agriculture.gov.au accessed on 6 March 2023 and used under the creative commons license CC BY 4.0. http://doi.org/10.25814/aqjw-rq15.

Other factors that determine if cover crops will successfully establish in Australian cropping systems include the existing weed pressures in the cropping system. If existing weed pressures are too high, weeds may dominate and quickly suppress small-seeded cover crops, which require sufficient time to germinate and establish. Some cover crops are not able to be easily established without an herbicide application to effectively control existing weed populations. In addition to moisture availability, the actual use of soil moisture by established cover crops may impact biomass accumulation over time, particularly if soil moisture is limited.

The choice of cover crop species is frequently impacted by cropping systems, regional adaptation, and the presence of local pests. In addition, the cost of establishment of the cover crop is an exceptionally important factor to consider as well. Small-seeded covers may prove difficult to seed, particularly if precision seeding is required on a large scale. Seed costs may also prove to be high, depending on availability. Frequently, once demand for a particular or newly established cover is met, private industry, or in this case, Australian seed companies, will respond by producing seed on a commercial scale to allow for market expansion. A good example of this is the winter annual legume covers, which can also be used as cover or rotational pasture crops [35].

Recently, mixtures of multiple species planted as cover crops have also been attempted to enhance overall establishment, improve total biomass production, and select mixtures in which one or even several species respond favourably to variable climatic conditions or extremes [36]. Research to assess the impact of binary and multiple species mixtures

on biomass accumulation and productivity has been conducted more globally and is now also underway by our research team in Australia as well. Useful mixtures of winter cover crops include oats and vetch (*Vicia* spp.), oats and field peas (*Pisum sativum*), or oats alone. While planting cover crop mixtures may result in increased biomass accumulation, it is difficult to predict productivity and soil moisture usage unless replicated experimentation is conducted in various locations and soil types. In our recent experiments, certain mixtures of cover crops tended to outperform monocultures in terms of biomass produced and competitiveness against weeds. Useful mixtures included oats plus vetch, field pea, or also tillage radish.

Recently, Australian producers have turned to a selection of diverse legume cover crops for production on sandy or less fertile soils. This is because legumes, even if seeded for short periods of time, can successfully fix significant N over time, thereby contributing to improved soil N status, while potentially altering the soil microbiome, thereby creating positive impacts on N cycling, soil health, and the subsequent cropping system [8]. However, it can still be challenging to locate legume species that can successfully adapt to the diverse climatic conditions and soil types encountered across Australia. Frequently, the availability of desirable legume species can be limited by a lack of choice, quantity, or cost.

The selection of a cover crop can also impact soil and above-ground pest prevalence. Liu et al. [37] describe a series of experiments with cover crop residues (niger seed, phacelia, rapeseed, radish, vetch, black oats, and buckwheat) on seedlings of selected cash crops (asparagus, carrot, onion, and sugar beet) in sandy soils with a history of soil-borne fungal diseases (*Fusarium* and *Rhizoctonia*). Tolerance of sugar beet seedlings to *Rhizoctonia solani*, a common soil-borne pathogen, was associated with increasing amounts of crop residue, resulting in an increase in the frequency of root-associated *Oxalobacteraceae*, *Bacillaceae*, and *Mortierellaceae*. Similar cover crop residues also increased the *Fusarium*-induced failure of asparagus seed germination. In these examples, the presence of cover crop residue impacted the soil microbiome community structure, resulting in the protection of certain cash crops while altering the successful establishment of others. In the subsequent sections of this review, we utilise specific case studies to illustrate how diverse cover crops are used in Australian production systems and the benefits and challenges associated with their integration into the rotational schemes currently employed.

## 3. Inclusion of Cover Crops in Mediterranean Conditions in Sandy, Low Organic Matter, and Sodic Soils

In Mediterranean regions, intensive cropping practices have significantly and negatively impacted soil health and water retention, particularly in sandy loam soils [38]. Summer cropping intensive regions in Central Europe have benefited from the use of winter cover crops that accumulate high levels of biomass, including annual ryegrass species, which, when used in combination with leguminous crops such as various vetch species, reduce the weed burden and improve soil N and organic carbon levels in a sequential crop rotation regime [39]. However, the balance between the available soil moisture for subsequent crops and the improvement of soil tilth by the cover crop is highly dependent on the prevalent climatic conditions, including soil moisture availability, and the sowing windows employed by the grower.

In the early decades of the 2000s, research efforts by the Western Australian Department of Agriculture, the New South Wales Department of Primary Industries, Murdoch University, and Charles Sturt University identified several leguminous annual pasture legumes suitable for inclusion in Mediterranean-type environments in the region, including biserrula and serradella, *Biserrula pelenicus* and *Ornithopus sativus*, respectively [40]. Although primarily introduced to Australia to bridge the feed gap for grazing livestock in the warm summer months, several of these species were also found to be highly competitive against weeds and provided groundcover throughout the growing season when managed appropriately, serving the role of a cover crop [23,41,42]. Additionally, the hard-seeded nature of these species creates some resilience in growing seasons with low rainfall, which

allows their utilisation as an "on-demand" break crop solution and the reappearance of these species in broadacre crops several years following initial establishment [35].

In dryland trials conducted in Western Australia, a wheat monoculture was the most profitable among the rotations over 12 years at Cunderdin, 156 kilometres east of Perth. This was largely due to the relatively stable wheat income, even in dry years, and the less beneficial or cost-effective impact of growing cover crops in the first three years of the trial in cereal and other more diverse rotations. Cover crops in the rotation provided limited to no grain income and little positive effect on subsequent crop yield [43]. The 'maximum carbon input' cereal rotation over 12 years had a mean gross margin of $302/ha; the 'maximum diversity rotation' (cereal, legumes, and brassica with maximum residue retention) had a mean gross margin of $292/ha; the 'wheat monoculture rotation' had a mean gross margin of $325/ha; and finally, the 'maximum profit rotation' (following the local practice of cereal, legumes, or fallow with lower residue retention) returned $301/ha [43]. Whether continuous wheat crops (the best in the trial) can be sustained without eventual damaging pest incursions, however, is questionable depending on climate and location, which is a very important consideration. In contrast, the use of such diverse legumes in the Riverina region of south-eastern NSW has offered potential for mixed croppers wishing to sustain grazing livestock longer in late spring with the high-protein feeds on offer [23,35,44].

Rose et al. [45] present a thorough analysis of the prospects for summer cover crops in Southern Australia, south of 32° S; a line roughly through Muswellbrook, Dubbo, and Broken Hill (NSW) to Ceduna and the cropping zones of South Australia and Western Australia. Their conclusions are compatible with our sources regarding the semi-arid Mediterranean environments. With wet, mild winters and hot, dry summers, these climates are generally hostile to the use of cover crops in broadacre farming but occasionally permit opportunistic use when sufficient soil water is present. For example, where (i) significant erosion risk appears due to reduced soil cover following winter crop failure, (ii) persistent herbicide-resistant weeds in winter crops "require a chemical spring fallow and summer cover/forage crop, which could be grazed", (iii) soil profiles full of water at harvest of the winter crop and a wet seasonal outlook provide opportunities for cover crop growth without weed competition, and (iv) sandy soil profiles with low water holding capacity combined with reliable winter rainfall such that winter crop yields do not rely on summer rainfall. Remaining unknowns are currently centred around "the values of more diverse cover crop species to soil health in cereal-brassica or cereal-brassica-legume cash crop rotations" [45].

## 4. Soil Moisture Conservation through the Inclusion of Summer and Winter Cover Crops

A higher proportion of summer rainfall observed in southern Australia in recent years has prompted renewed focus on the capture and storage of soil moisture collected over the summer fallow, which spans approximately four to six months preceding the subsequent autumn and winter-sown crops [46]. In central NSW locations where soils have greater water holding capacities and experience variable rainfall over spring, conserved summer rainfall in the soil profile can contribute up to an additional 2 t/ha of wheat yield in simulations [47]. Reliable summer rainfall capture and retention is primarily reliant on the capacity of rainfall to penetrate the upper soil layers and collect below the evaporation zone. However, maintaining a stringent summer weed control program is paramount to reducing stored water loss through evapotranspiration. However, such chemical fallows provide opportunities for herbicide-resistant summer weeds to establish and use stored soil moisture [48]. In southern Queensland, where annual rainfall is mostly summer dominant, grey cracking vertosols benefit from reduced erosion by additional groundcover from the establishment of short-season summer grasses such as millet (*Setaria* spp.), which provides an additional 20 mm of plant-available water to a wheat crop in the subsequent winter growing season [49]. Importantly, this late-terminated cover crop provided better ground cover, resulting in more uniform establishment of the wheat crop with a yield increase of

two-fold (2.8 t/ha) and a net economic benefit of \$280/ha when compared to a chemical fallow.

Furthermore, winter cover crops can provide vital groundcover to conserve soil water, preceding a high-value summer crop (e.g., cotton) [50–52]. In summer cropping systems, a large proportion of rainfall and irrigated water is lost through evaporation, prompting a need for improved groundcover and penetration of water deep into the soil profile [53]. Field evaluations on the inclusion of winter-sown cover crops in southern Queensland have shown that late-terminated barley provided excellent groundcover [29]. However, the longer growing period required to ensure increased cover crop groundcover and biomass resulted in greater soil water use by the cover crop (120 mm when compared to 70 mm for mid-termination). It is important to note that the improved groundcover continued to provide soil moisture savings over the cotton growing season, accumulating soil water similar to a cover crop-free treatment. The cover crop treatments also provided an additional 3 bales/ha of cotton when compared to the control, providing strong evidence that the soil water savings over the fallow and growth of the cover crop significantly benefited the performance of the cotton crop. Incorporating vetch with barley (*Hordeum sativum*) as a cover crop mixture also used a similar quantity of water to a sole barley cover crop while providing additional N inputs to the soil.

In a survey of Southern Queensland and Northern NSW growers of dryland cotton, Walker et al. [54] reported the following: "In general, weed management in dryland cotton involves controlling weeds in previous fallows and rotational crops using selective herbicides, as well as inter-row cultivation, inter-row spraying of non-selective herbicides with a shielded sprayer, and manual chipping". The use of cover crops to control weeds and conserve soil water in advance of planting dryland cotton was not described. Bange et al. [55] reported that some Queensland farmers were using seasonal climate forecasts in conjunction with systems analysis to reduce the impacts of some climate risks [56]. By identifying climate-related decisions considered to influence the overall farm operation in either economic or environmental terms, these farmers gained a better understanding of the system's vulnerability and aimed to incorporate climate risk management into their operations. Examples of when a forecast is 'likely to be drier than normal' include maximising no-till areas (water conservation), applying some N fertiliser early to allow planting on stored soil moisture but splitting the application, allowing more to be applied later if a good season eventuates, and planting most wheat later than normal to reduce frost risk and increase row spacing. In seasons forecasted as 'likely to be wetter than normal', management options include sowing wheat earlier, applying N to a wheat cover crop grown on a dry profile after cotton (normally not expected to produce a harvestable yield but weed-smothering groundcover), and applying fungicides to wheat crops to minimise leaf diseases.

Today, the use of seasonal weather forecasts has 'caught on' in Australia and proven valuable in a number of sectors. Among six commodities considered in a recent economic analysis, cotton promised the highest payoffs from seasonal weather forecasts [57]. Weed management practices in cotton systems in New South Wales and Queensland (Australia) changed from using residual herbicides and cultivation to permanent beds, reduced cultivation, and a reliance on glyphosate [58]. The introduction of glyphosate-resistant cotton in 2000 encouraged greater reliance on this herbicide. Glyphosate-resistant cotton allowed up to four in-crop applications of glyphosate; combined with a pre-planting glyphosate application, up to five applications may occur in the cotton cropping season. Popular reliance on this easy and relatively inexpensive weed control method inadvertently selected for no less than 10 glyphosate-tolerant species [59], two of which have evolved into glyphosate-resistant weeds (*Sonchus oleraceus*, "annual sowthistle", and *Echinochloa colona*, "awnless barnyard grass") [60,61].

In 2017, Luo et al. [62] published a bioeconomic simulation study about "the economics and the risk of adaptation options in the Australian cotton industry", given the expected future temperatures and rainfall regimes projected with climate change by 2030. This study looked at irrigated crops at three locations: Dalby (Queensland), Narrabri, and Hillston

(New South Wales), for irrigated crops at all sites and dryland crops only at the Dalby and Narrabri sites. Adaptations considered changes to earlier and later planting dates, row configurations, irrigation triggers, and the inclusion of one fallow season with each of the two cotton seasons, compared with continuous cotton. Two irrigation triggers were considered for irrigated cotton: low (50 mm) and normal (70 mm) available soil water, below which an irrigation event is triggered. Irrigated cotton at Hillston was projected to gain less than 10% in gross margins from early planting; less than 5% at Narrabri; but not at all at Dalby, where Dalby gross margins at the normal planting date were expected to improve more than 10% with climate change.

*CottonInfo* is the Australian cotton industry's joint extension program on best practices. Recently, a special issue of the *CottonInfo* newsletter (2018) [63] focused on cover cropping strategies. Six farmers across Australia's cotton regions described how and why they have integrated cover crops. These strategies include use of: (A) standing wheat straw or remnants of a previous corn crop retained for winter rain infiltration and minimizing soil erosion and weeds before sowing cotton into the crop residue; (B) planting cotton into a dense sacrifice cereal crop to prevent soil crusting and water runoff while protecting young cotton plants from wind and sand blasting; (C) using vetch cover crops as green manure, terminated prior to sowing cotton, with the intention to build soil organic matter and manage weeds; (D) a general rotation of cotton planted into standing wheat stubble: cotton harvest followed by fallow and wheat before the next cotton crop to retain soil moisture and structure while building organic matter; (E) low-input wheat as a repair crop following cotton to fix compaction, increase organic matter, and improve biological activity in the soil; (F) a general rotation of dryland cotton, followed by wheat, a pulse crop, wheat, and back into cotton to allow flexible production of crops to maturity or early termination prior to seed set, depending on seasonal weather expectations.

One producer (at Boggabilla, near the NSW-Queensland border) expressed a thought that must be on the minds of many: "My burning question about cover cropping is the impact on yield. This is a long-term approach for us, but I am interested to know the time period in which we might start seeing the impact of cover cropping on yield. And, if yield increases, how will we know if it is cover cropping that has made the biggest contribution?" Most of the interviewed farmers pointed to well-reasoned biological sustainability justifications for their cover cropping efforts [63]. The next question must be the nature of the economic payoffs needed to cover cropping over time, given the highly variable weather conditions from year to year.

Irrigated cotton systems require tillage for stubble management and dealing with pest (weed and insect) resistance. There has been a move away from burning cotton stubble in favour of incorporation of mulched crop residues into the surface soil, timing the tillage to reduce the number of overwintering insects. This tillage is a mandatory requirement when using non-transgenic cultivars that lack insect pest resistance. Generally, there is no irrigated cotton system available that relies on a 'single pass' tillage operation. Reduced tillage, in conjunction with cover crops or rotations with high residue crops, appears to offer increased productivity in cotton systems [64].

Cover crops are also grown to reduce long fallow periods in cotton cropping cycles, protect the soil from erosion, and reduce nutrient loss through erosion or leaching. However, incorporating cover crops as part of cotton rotations is difficult in highly capitalised, mechanised systems [65]. A better understanding is needed of soil and cotton yield improvements, water requirements, the cost of cover crops, and the impact of the cotton crop on nutrition uptake [64].

Cover crops can boost soil water storage and crop yields in Northern Queensland farming systems where up to 60% of rainfall is lost to evaporation and a further 5–20% is lost in runoff and deep drainage. Every 10 mm of extra stored soil water available to crops could increase dryland grain yields for growers by up to 150 kg/ha, with corresponding benefits for dryland cotton growers as well. When applying overhead irrigation, Erbacher et al. [66] observed increased stored water early in the cotton crop after cover crops, similar

to their fallow gains observed in dryland scenarios. The trial results showed a greater cotton yield following a range of cover crops, compared to the bare fallow. The researchers also observed the need for careful planning of cover crops prior to cotton crops, as cotton seeds planted on top of a cover crop row had poor seed-soil contact and did not establish, creating gaps in the plant stand and therefore reducing yield potential.

Charles and Taylor [67] summarised insights on cover crops, which are increasingly recommended in integrated approaches to controlling glyphosate-resistant Palmer amaranth and other troublesome weeds in cotton. Best Management Practices used in integrated weed management include: intensified crop rotations to disrupt recurring weed life cycles; conservation tillage (CT) systems to maximise cover crop residue biomass; delayed cotton planting to optimise benefits of increased soil temperatures and growing degree days promoting cotton growth; increased scouting to identify weed emergence or any other management problems early; timely herbicide applications for best efficacy; use of diversified herbicide chemistries (including pre-emergent or PRE herbicides) to reduce resistance pressure; and intermittent inversion tillage (strategic tillage) to bury weed seed beyond successful emergence depth [68–70].

After harvesting cotton or grain sorghum, a cover crop added in the winter growing season increased soil organic matter, recycled nutrients, prevented erosion, and suppressed weeds. In south-east Texas, where cotton is planted into cover crop residues, a recent set of Texas A&M trials showed that oats produced the highest biomass and offered the best weed suppression in early-terminated crops. Oats also performed well in the late-termination group, placing second in both biomass and weed suppression [71].

## 5. The Impact of Cover Crops Established as Living Mulches in Viticultural Crops

In areas with a Mediterranean climate, ensuring sufficient water for the crop is an important concern. Unlike many field and fruit crops, however, which require high water availability, vineyards generally require moderate water stress to produce the grape quality necessary for wine production [72,73]. Therefore, water management in vineyards must avoid two excesses. If water resources are unlimited, vegetative development can be luxuriant but associated with poor grape maturation and a high risk of fungal attacks, requiring repeated trimming and topping. Excessive water stress markedly restricts leaf growth, particularly if it occurs before flowering, affecting the net assimilation rate and consequently yield and grape quality. The use of cover cropping is currently increasing in vineyards, but its development remains slow in Mediterranean regions because of the possibility of severe competition for resources. However, recent studies on intercropping have shown that in some situations, water stress may not be greater than that prevailing in bare soil vineyards [74].

A cover crop may be a single species, such as a winter cereal sown to protect the soil in a newly planted vineyard. The cover crop may also comprise several grass, herb, and legume species to provide diversity. The species may be winter- or summer-growing annuals. The energy of the falling rain is dissipated by an intercepting cover crop, which also slows down runoff. Without the protection of a cover crop, weakly structured soil may not only erode but, on drying, may also form a surface seal or crust that inhibits infiltration and hence worsens the erosion potential [75]. Abad et al. [76] have published a systematic world literature review of cover crops in viticulture with respect to: (1) implications on soil characteristics and biodiversity; and (2) implications on vineyard agronomic performance.

Geldenhuys et al. [77] showed that maintenance of dense and diverse cover crops is a strategy to enhance vineyard arthropod diversity (spiders, beetles, and true bugs) where other management practices are already sensitively implemented, thereby complying with the Integrated Production of Wine (IPW) certification scheme (http://www.ipw.co.za; accessed 6 March 2023), allowing South Africa to provide a visual guarantee, not only for the integrity of origin and/or vintage year and/or cultivar, but also for sustainable production and traceability. This publication highlights that minimal adjustments to management practices can greatly benefit farmland biodiversity conservation and are in keeping with

the ethos of the concept of a biosphere reserve. In addition, vineyard cover crop diversity can promote important service-providing arthropods in vineyards, such as natural enemies of pests and species of conservation concern, including wild bees and specialist butterflies. Most frequently in evidence are cover crops in the inter-row areas, between the vines, with under-vine areas kept free of undergrowth by cultivation or herbicide sprays. This has raised the subject of finding under-vine plant species that could suppress pests without compromising vine yield or quality. An underlying justification for this research has been to minimise herbicide use in vineyards, due to the concern in the European Union over banning imports of foodstuffs produced with glyphosate [33].

Commercial perennial agriculture is unfortunately prone to declining productivity due to negative plant-soil feedback. An alternative to costly and environmentally harmful conventional treatments such as soil fumigation could be to manipulate soil microbial diversity through the careful selection and management of cover crop mixtures. Although cover crops are already used in these systems for other reasons, their capacity to influence the soil biota is often untapped [78]. Frequent tillage, herbicide use, and copper fungicides have been thought to harm populations of beneficial microbes and, in some cases, contribute to greater crop decline. Non-crop vegetation management could be a viable and cost-effective means of minimizing crop decline in perennial monocultures but requires more direct experimental investigation in perennial agroecosystems [78]. Various species considered weeds in broad-acre agriculture may not be out of place in a vineyard cover crop, provided they have no specific adverse effects on vine growth. In vineyards, perennial cover crops that grow throughout the year may also be suitable if soil moisture is adequate [75].

Some grape growers, or vignerons, allow the cover crop to grow into the vine row when they are not concerned about competition with the vines for water and nutrients. Indeed, on high-yielding sites, a cover crop can be useful in competing with the vines for water and nutrients, especially N, and thereby restricting vigour, for the sake of improved wine quality [33]. Weed suppression is often more effective with perennial cover crops than with annuals, especially when the annual species die off in summer. Efficacy also depends on the management of the cover crop, such as whether it is mowed regularly, whether it extends under the vine, and the vigour of the species present, particularly the grasses. In vineyards, the SOC content will slowly decrease over time when the mid-rows are cultivated because the returns of carbon to the soil are small. However, when there is a permanent cover crop or when compost is used, SOC may slowly increase over time. SOC sequestration with the introduction of cover crops has also proven to be highly efficient [79].

Cover crop presence can also impact vine growth, grape production, and quality, as seen in Canada [80]. Four treatments (three mixtures: (i) oats + peas + hairy vetch (OPV), (ii) oats mixed with red clover (ORCl), (iii) timothy + alsike + red clover (TM), and (iv) control with no cover crop (CONT)) were the basis of a trial in a vineyard (*Vitis vinifera*) in Nova Scotia, in eastern Canada, a fast-growing wine production area located in the mid-temperate zone with a humid continental climate and given exposure to the warm Gulf Stream. Best-yielding was the ORCl treatment at 1.46 kg/vine, followed by TM at 1.31 kg/vine, while the Brix (sugar) levels were 13.71 and 15.73, respectively. This suggests the TM treatment used more soil water, slightly stressing the vines and boosting the quality of the grapes. In 2011 and 2012, the trial site at $44°49'$ north latitude, near Halifax, recorded May–October growing season rainfalls (GSR) of 871 mm and 597 mm, respectively, and mean growing season temperatures (GST) in the same years were 14.6 and 16.3 °C.

By way of comparison, the Nova Scotia trial site was wetter and warmer than Tasmania's south-east wine region in 1997–2017, where October-April GSR was 454 mm with a GST of 13.8 °C at Hobart, $42°52'$ south latitude [81]. Nevertheless, the Nova Scotia and Southeast Tasmania regions both produce high-quality wines made from Chardonnay and Pinot Noir varieties, as well as others [82,83].

## 6. Cost-Benefit Analyses of Using Cover Crops in Australia—An Economic Perspective

The harsh Australian climate creates frequent and large yearly price and weather variations from year to year, in contrast to those observed in other regions of the world [56]. This means a given production technique may yield very different results over time—one year is excellent and the next is not. Further variance in economic outcomes arises due to the low correlation between international commodity prices and Australian local weather, sometimes combining poor yields with poor global prices, and sometimes high yields with high global prices. Successful producers are also business savvy, and one of the first questions asked by farmers and crop consultants considering a transition to a novel practice is, "What is the business case for adopting such a practice?" [84]. Improved cash crop yields, due to a change in production practice over one or more years of field trials, are often a key to acceptance by producers. The other key aspect of interest to producers includes the evaluation of gross margins (i.e., taking that increase in crop yield times the sale price/kg, minus the increase in production costs ($/ha), to define gross margins). Given that prices, weather, and cash crop yields typically vary from year to year, it is most useful to estimate the range of gross margin outcomes over time compared with those of the current practice (current prices). Single-point estimates of gross margins often leave many questions unanswered. However, some Australian examples of cover cropping comparisons of gross margin distributions over time, which clarify these differences, can be described.

Bell et al. [85] reported their analysis of a large set of trials aimed at comparing complex farming system mixes (baseline; higher nutrient supply; higher legume; higher crop diversity; crop diversity + nutrient; higher legume + diversity; and lower crop intensity) applied in four classes (mixed opportunity; higher intensity; summer; and winter) at each of seven geographically separated sites (with representative soils) in Queensland and Northern New South Wales from April 2015 to November 2019. Standardised production costs for seed, fertilisers, sprays, etc. at each location were applied to gross margin budgets for each farming system mix. Farm gate prices for crops at each location were based on 10-years of port prices, adjusted to allow for transport, grading, or bagging costs or losses.

Each of the cropping systems gave gross margin results over the 10 price years traced as a cumulative distribution curve. At a given location, the right-most gross margin curve indicates the highest gross margin. Common among the seven locations is the bold, solid black curve showing the overall performance of the recommended baseline practice; notice that its results are often the best or second-best among the options. This finding suggests that treatment results are largely due to variations in site conditions. Only at two locations (Emerald and Trangie grey-red soil) did a higher nutrient supply give better results than the baseline treatment. The higher crop intensity option produced gross margins comparable to the baseline only at Narrabri; elsewhere it was among the poorest options. At all locations tested, higher crop diversity produced poorer results than the baseline treatment. Zull et al. [86] provide further clarifications of the above results.

Another recent study from Queensland, Erbacher and Lawrence [87], reported results of cover crop trials in 2018–2020 from a cluster of three locations in southeast southwest Queensland (Yagaburne, Goondiwindi, and Billa Billa) and Croppa Creek (northeast NSW), each with different soil conditions and rainfalls, also producing mixed results. These results further demonstrated the wide temporal and spatial variations in productivity gains with cover cropping; cover not only protects the soil from erosion but also increases water infiltration and plant availability of water, often but not always raising subsequent crop yields. Whish et al. [88] earlier quantified the risks implied for water used by a cover crop to 'rob' the subsequent cash crop of water. This ambiguous word 'rob' points to the fact that outcome is logically dependent on conditions associated with a particular cropping season and type and duration of cover crop.

Nordblom et al. [33] most recently focused on bioeconomic comparisons of under-vine cover crops in the vineyards of four South Australian wine districts. The control treatment at each trial site was a typical herbicide spray, and each site had a straw mulch treatment

and two treatments of different pasture legumes or grasses (the Riverland district was an exception with herbicide control and three pasture species treatments). District-specific treatment costs, grape yields, and prices over the 2006–2017 period allowed calculation of probabilistic relations of yields and prices (as jointly distributed random variates) for distributions of gross margins specific to each treatment in each district. Penfold et al. [32] provide results for ten treatments at each site and all the biological information from the field trials, on which the economic analysis below is based.

In the Barossa district, the herbicide treatment gave the lowest gross margins (Figure 2a), while in the Riverland district, the herbicide treatment generated the highest gross margins. Results were also mixed in the other two districts. Figure 2b shows the value of going beyond average gross margins; in Barossa, the perennial ryegrass (*Lolium perenne*)/medic (*Medicago* spp.) under-vine treatment allowed the vineyard to remain financially viable even under the highest level of opening debt. Furthermore, the inclusion of perennial cover crop species as under-vine covers resulted in improved fruit quality, as assessed by sensory analysis. This is potentially due to the greater abundance of arbuscular mycorrhizal fungi when compared to control soils in the absence of covers, which have been associated with improved wine quality [72].

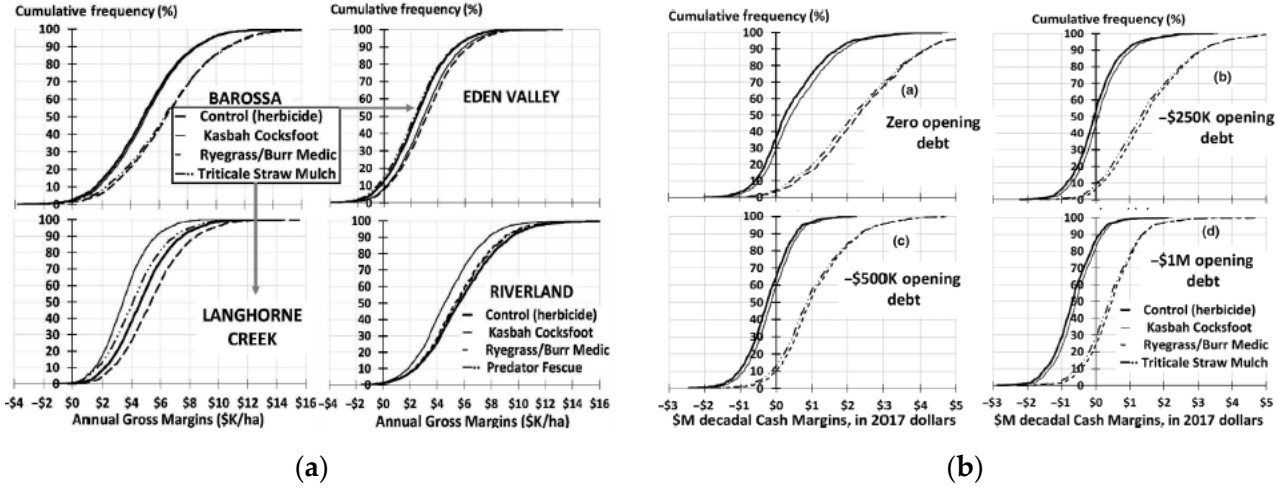

**Figure 2.** Cover crop treatment gross margins for four South Australian wine districts. (**a**) Treatment decadal cash margins (risk profiles) at four opening debt levels as specified on graph b for a 50-ha Barossa vineyard (**b**) (Adapted from Nordblom et al. [33]).

Another economic analysis of cover crop options conducted by this Australian team examines the question: Does establishing lucerne under a cover crop increase farm financial risk [89]? The case examined was for rainfed farms in south-eastern Australia, which often combine annual cropping and perennial pasture phases with grazing sheep enterprises. Such diversity assists in managing diseases, pests, and plant nutrition while stabilising income in the face of wide, uncorrelated variations in international commodity prices and local weather. An actuarial accounting approach was used to capture the above contexts and render financial risk profiles in the form of distributions of decadal cash balances for a representative 1000-ha farm at Coolamon, New South Wales, Australia.

Wheat cover crops have been shown to assist lucerne establishment, performing the role of a nurse crop while also supporting the subsequent harvest of wheat, followed by the successful grazing potential of lucerne pasture for the following several years (generally three to four; Figure 3). However, the late emergence and establishment of lucerne due to adverse weather conditions or moisture limitations may reduce its capacity to compete with the wheat crop for resources. For the case at hand in the Coolamon area, over time, sowing lucerne alone appears to be the better practice, regardless of the observation that desired results may occur frequently but not in every growing season. This could be associated

with the presence of residue creating negative feedback when establishing lucerne due to allelopathic interactions with degrading wheat residues, physical suppression due to residue density, pathogens, or other pests associated with the residue.

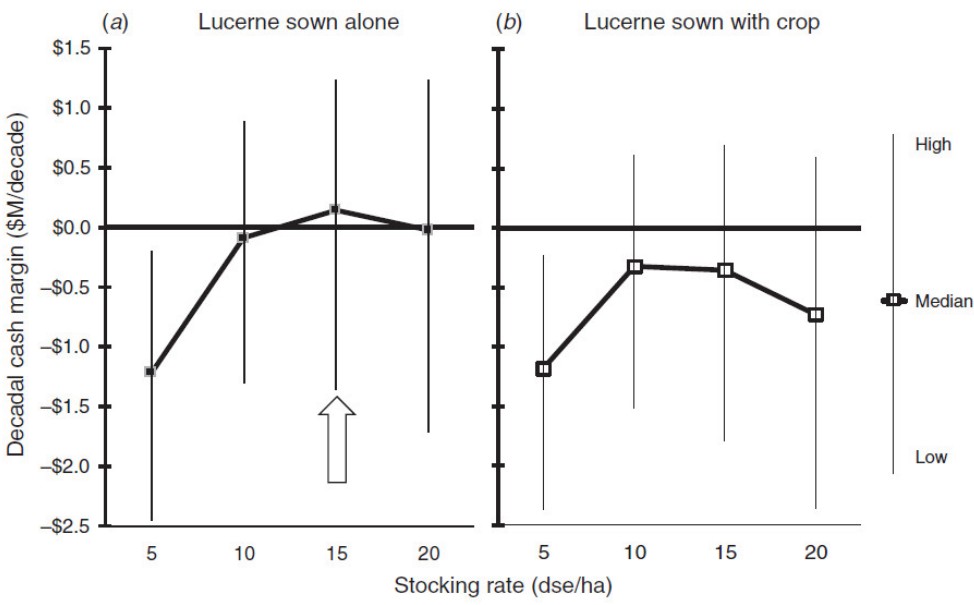

**Figure 3.** Effects of lucerne establishment methods, either (**a**) sown alone or (**b**) with a cover crop, under different stocking rates, on the distributions of decadal cash margins, given 80% opening equity on a farm with 400 ha of sown pasture in phased rotation with 500 ha of crops in Nordblom et al. 2017 [89]). The white arrow refers to the optimal combination of lucerne sown alone and stocking rate (15 density of sheep equivalents dse/ha).

A comparative study of cotton and grain farming systems in south-eastern Queensland and northern New South Wales was performed by Zull et al. [86], based on researcher-managed field trials. The scientifically designed plans for controlled, paired field experiments, developed in cooperation with producers, were properly measured and analysed by researchers. The results are valuable from three points of view: (1) comparing baseline recommended treatments and alternative treatment mixes (some including cover crops) specific to eight locations with different soils and microclimates; (2) repeating these mixes at each site over five seasons to capture natural variations in growth conditions; and summarising the physical results over time; and (3) translating the results into gross margin outcomes summarised as cumulative frequency distributions of gross margins ($/ha), from lowest to highest for each treatment at a given location, easily understood by growers. Full details of these trials are given in Zull et al. [86] as an extension document and summarised more concisely by Zull et al. [90].

Zulauf and Brown [91] describe a nation-wide summary of cover crops in the US, indicating adoption by only 3–7% of farms in the Midwestern states. Unfortunately, no such summary appears available for cover crops in Australia, where no-till conservation farming is quite common and cover cropping relatively rare.

Considerable advancement, in the form of modern extension videos and print documentation on cover cropping, is apparent in the work of the 'Sustainable Agriculture Research and Education' (SARE) program under cooperative agreements with the National Institute of Food and Agriculture, USDA, the University of Maryland, and the University of Vermont [92]. Their message is that cover crops can provide opportunities to 'Improve Your Bottom Line in Row Crops'. As Rose et al. [45] have delimited the conditions in which cover crops might be most beneficial in southern Australia, Myers et al. [92] have elucidated "seven common management situations for commodity farmers that can affect how quickly they receive a positive net return from cover crops in the Midwest: (1) where

herbicide-resistant weeds are a problem; (2) where cover crops are grazed; (3) where soil compaction is an issue; (4) where cover crops are used to speed up and ease the transition to no-till; (5) where soil moisture is at a deficit or irrigation is needed; (6) where fertiliser costs are high or manure nutrients need to be sequestered; and (7) where incentive payments from the government are received for using cover crops". Acharya et al. [93], for example, discuss the role of government subsidies in allowing cover crops in the Southern High Plains of Oklahoma, Texas, and New Mexico in the US.

The latter motive continues to be absent in Australia [94]: "Unlike in other developed countries, there is generally no expectation in Australia and New Zealand that farmers will be fully compensated for the costs of complying with agri-environmental policies." However, there appear to be options for collective agreements that specify that recognised environmentally sustainable practices, such as cover crops and no-till conservation farming, can be certified to serve as a legitimate marketing 'trademark' or stamp in which consumers can have confidence. This may serve best where products are marketed in small packages (coffee, almonds, cherries, etc.) rather than as bulk commodities. In the latter case, though, a food processing industry may offer contracts for certifiable lots of bulk cereal grain or oil seed from farmers who can guarantee the required qualities in their production process. That 'stamp of quality' may then be displayed on the industry's products, which consumers can learn to favour.

SARE [92] has produced the 3rd edition of a compendium providing exhaustive coverage of the main cover crop species (seven non-legumes, three brassicas, and eleven legumes) with respect to their individual suitability as an N source, a soil builder, an erosion fighter, a subsoil loosener, a weed fighter, or a pest fighter in each of fourteen specified regions across the USA, in roles such as providing lasting residue, long-duration ground cover, harvest value, or cash crop interseeded as a living mulch. In addition, the report covers cultural traits and tolerances, planting requirements, and their potential advantages and disadvantages. In summary, the report suggests cover crop selection is dependent on region and climate, and value can be provided by successful establishment and use patterns in terms of erosion prevention, weed suppression, pest reduction, and N provision or fixation.

## 7. Summary

In southern Australia, the yield benefits of cash crops following the incorporation of cover crops differed between farming systems and were impacted by the local climate and soil type within the same farming system. This suggests a need for further performance evaluation of specific cover crop species and their ecosystem services at a regional level over multiple years to better inform growers on their suitability. It is also apparent that broadacre field crops are likely to continue benefiting from no-till conservation techniques and emerging innovations in automation [9], but perhaps less significantly from winter cover crop inclusion in the absence of livestock. Innovative means for selecting crops adapted to regional Australian rainfed conditions and the use of precision planting to enhance their establishment under optimal field conditions and timings will certainly assist in the uptake and use of successful cover crop genotypes across southern Australia.

**Author Contributions:** Conceptualization, L.A.W., S.G. and T.N.; methodology, L.A.W., S.G., A.E. and T.N.; formal analysis, T.N. and S.G.; investigation, L.A.W., S.G. and T.N.; resources, L.A.W. and S.G.; writing—original draft preparation, S.G., T.N. and L.A.W.; writing—review and editing, S.G., T.N., A.E. and L.A.W.; funding acquisition, L.A.W. and S.G. All authors have read and agreed to the published version of the manuscript.

**Funding:** This research was funded by the Grains Research and Development Corporation, GRDC, grant number US 000084.

**Data Availability Statement:** Data used in this study is available from the relevant cited articles.

**Acknowledgments:** We acknowledge the efforts of those who have worked before us by breeding, selecting, and cultivar-trialing various genotypes of potential cover crops for Australian conditions,

particularly Angelo Loi, Belinda Hackney, and Brad Nutt. We also acknowledge the efforts of our research cooperators, especially Michael Walsh, who led and administered GRDC research project US 000084 through the University of Sydney and was further supported by Jason Emms of the GRDC in Crop Protection.

**Conflicts of Interest:** The authors declare no conflict of interest.

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
