# Peer review of "Opportunities and Challenges for Cover Cropping in Sustainable Agriculture Systems in Southern Australia"

_agriculture, doi:10.3390/agriculture13030688_

Round 1

Reviewer 1 Report

Dear editor,

The review manuscript focused on the Opportunities and Challenges for Cover Cropping in Sustainable Agriculture Systems in Southern Australia. This review is suitable for publishing in the agriculture journal but the authors should modify some corrections. Therefore, my final recommendation is "Minor revision".

- Your abstract should clearly state the essence of the problem you are addressing, what you did and what you found and recommend. That will help a prospective reader of the abstract to decide if they wish to read the entire article.

- In the first paragraph of introduction, the authors should be noted the negative impacts of herbicides on the agricultural ecosystems and human health.

- Also, I recommend to authors for adding the herbicides using amount in this country and the contribution of these chemical inputs using in increasing production costs.

- The linkage between introduction paragraphs is missed.

- L44: ‘tilth’ or ‘health’?

- 92-101: If you have statistics about cultivation area of cover crops in Australia, it is better to present it in a table.

- Based on previously published articles, please add statistics on the amount of damage caused by weeds to agricultural plants in a table to the article.

- Please add one table for effect of winter or summer cover crops on the weed suppression in Australia.

- Figure 2: The image quality should be increased.

- In summary section, what suggestions do you have for future research in this field?

Author Response

The authors thank the reviewer for his valuable feedback. We have made necessary changes where applicable, with responses to specific comments provided in tabular form. Thank you.

Reviewer 2 Report

Dear Authors,
The review paper needs rigorous effort to improve its quality as suggested below:
1. The review is appearing as location specific problem and hence ascertaining for worldwide readers is difficult. If the location name is deleted from the title, it can be broader for readers.
2. Abstract: Concise the abstract. Delete unnecessary statements and rewrite according to the format as mentioned in the attached reviewed manuscript.
3. Use abbreviations properly in the entire manuscript.
4. Add some photographs of cover crops found in the study area.
5. On reading the entire manuscript several grammatical and unnecessary statements were found, go through the whole manuscript carefully and correct/delete those statements.
6. Add some meta-analysis data/references (tables) to improve the quality of review paper
6. Summary is not written well, need to refine.
Overall, improve the manuscript

Author Response

The authors thank the in-depth feedback provided by reviewer 2. We have made extensive changes to the abstract and conclusion as well as some modifications to the body of text to emphasize the limitations and opportunities for cover cropping in southern Australia, specifically. 

Thank you

Round 2

Reviewer 2 Report

.